# The History of Clinical Islet Transplantation in Japan

**DOI:** 10.3390/jcm11061645

**Published:** 2022-03-16

**Authors:** Taihei Ito, Takashi Kenmochi, Kei Kurihara, Naohiro Aida

**Affiliations:** Department of Transplantation and Regenerative Medicine, School of Medicine, Fujita Health University, Dengakugakubo 1-98, Kutsukakecho, Toyoake 470-1192, Japan; kenmochi@fujita-hu.ac.jp (T.K.); kurihara@fujita-hu.ac.jp (K.K.); n-aida@fujita-hu.ac.jp (N.A.)

**Keywords:** islet transplantation, type 1 diabetes, national health insurance

## Abstract

Islet transplantation shows the promise of being capable of relieving glucose instability and improving QOL of patients with type 1 diabetes that cannot be controlled due to severe hypoglycemia unawareness. In Japan, following the first human islet isolation from a donor after cardiac death in 2003 and the first clinical islet transplantation in 2004, islet transplantation was performed for the improvement of type 1 diabetes as a single-center trial in several centers. Although it was discontinued due to the possibility of contamination of collagenase by bovine brain component in 2007, the phase II clinical trial of islet transplantation started using ATG induction and a TNF-α inhibition protocol in 2012. The primary endpoints of this trial were the proportion of patients with HbA1c < 7.4% and freedom from severe hypoglycemic events at one year after the first islet cell infusion. In an interim analysis, this endpoint was achieved in 75% of cases. In April 2020, clinical islet transplantation was finally covered by health insurance in Japan, thanks to these outcomes. We herein introduce more than 20 years of history of clinical islet transplantation in Japan.

## 1. Introduction

Islet transplantation has been clinically applied in various countries as a cell therapy capable of recovering lost insulin secretory capacity. It is considered less invasive than pancreas transplantation. Lacy’s group developed remarkable methods for islet transplantation, including intraductal enzyme injection, density gradient isolation to improve islet yield and quality, and the intraportal infusion of islet grafts to be implanted to the liver [1], which is now the clinical gold standard. Subsequently, a semi-automatic method for islet isolation was developed by Ricordi et al. [2], and then the foundational technique of the current clinical islet transplantation method was almost completed. The world’s first clinical islet transplantation was reported by Najarian et al. in 1977 [3]. Since then, although a few successful cases of islet transplants have been reported [4], including auto-islet transplantation [5]; the rate of insulin independence after islet transplantation was only up to 11% until 2000 [6]; and significant breakthroughs, including an efficient immunosuppressive protocol for islet transplantation, have been awaited.

The results of clinical islet transplantation performed in Edmonton in 2000, which showed an insulin independence rate of 100% [7], were promising. However, unfortunately, patients suffering from type 1 diabetes did not show sufficient long-term results [8]. Subsequently, the use of thyroglobulin as induction therapy has dramatically improved clinical islet transplantation, including the long-term results [9,10,11,12,13], and has possibly surpassed the results of pancreas transplantation alone. Therefore, islet transplantation, which is less invasive than pancreas transplantation, should be considered an efficient treatment option for patients with type 1 diabetes; however, problems still remain, including the cost of islet isolation and post-transplant immunosuppressive therapy, as well as the fact that pancreas grafts have to be shared among patients waiting for pancreas transplantation and islet transplantation.

Transplantation medicine, which is established by receiving the donations of organs and tissues from patients after brain death or after cardiac death, is greatly influenced by the ethics that define death, legislation, and the social systems of each country. In Japan, a long-standing debate on the definition of brain death was held, and in 1997, a law on organ transplantation from brain-dead donors was finally enacted, and brain-dead organ transplantation was started under this law [14]. However, the severe shortage of donors has continued to be a serious problem, and many organ transplants from living donors were performed, including lung [15], liver [16], kidney [17], and pancreas transplantation [18].

On the other hand, clinical islet transplantation in Japan has long been carried out in clinical trials, in compliance with the guidelines set by the Japanese Society of Tissue Transplantation (JSTT) [19]; however, it is now obligatory to conduct clinical islet transplantation in accordance with the new regenerative medicine law that was enacted in 2013 [20]. Although clinical islet transplantation in Japan is classified as tissue transplantation and is considered to be outside the scope of the law on organ transplantation from brain-dead donors, the history of transplantation, including pancreas transplantation, is considered to be very deeply involved in the development of islet transplantation from the viewpoint of sharing pancreatic grafts.

Thanks to the successful outcomes of islet transplantation in the multicenter clinical trial started in 2012, in Japan, clinical islet transplantation finally received national medical insurance coverage in April 2020. This paper summarizes and reports on the progress in the more than 20 years from the beginning of clinical islet transplantation in Japan until it received national health insurance coverage. We hope that this paper will contribute to the implementation of clinical islet transplantation in other countries in the future.

## 2. Pancreas Transplantation in Japan

Since pancreas grafts are allocated for pancreas transplantation and islet transplantation, knowledge of the history of pancreas transplantation in Japan is important for understanding the history of islet transplantation. Therefore, this chapter summarizes Japanese pancreas transplantation.

Until the law on organ transplantation from brain-dead donors was enacted in 1997, there was no legislation for organ donation from brain-dead donors in Japan, so less than 200 cases of kidney transplantation from less than 100 donors after cardiac arrest were performed [21,22]. In this era, the Tokyo Women’s Medical University Group performed 11 cases of pancreas transplantation from donors after cardiac arrest [23,24]. Unfortunately, their report shows that although patient and kidney graft survival was relatively good, the pancreatic graft survival rate at 1, 3, and 5 years was 55%, 46%, and 36%, respectively.

In Japan, the law on organ transplantation from brain-dead donors was enacted in 1997, and brain-dead donor organ transplantation was carried out according to this law from that time. The first simultaneous pancreas and kidney transplantation in Japan under this legislation was performed by the Osaka University Group in 2000. Under this law, 86 cases of donation from brain-dead donors were carried out, and 62 cases of pancreas transplantation from brain-dead donors were performed in 13 years [25,26]. In this era, 27 cases of pancreas transplantation from living donors, including seven cases of ABO-incompatible transplantation, were also performed due to an insufficient number of brain-dead donors [18,27]. Since the shortage of donors in Japan was still serious, the law on organ transplantation from brain-dead donors was amended in 2009 and came into effect in 2010. In Japan, legal brain death is limited to human death only in the case of donation for organ transplantation. Before the amendment of the law, since brain-dead organ donation required the person to make their intention clear before brain death, for the first 10 years (approximately), only a few cases of brain-dead organ transplantation were performed each year. The primary modification of the law was that declaration of a person’s intent to donate before brain death is no longer required for brain-dead donation.

Since 2010, the number of brain-dead donor organ donations has gradually increased, reaching a record high of 98 cases in 2019 [28]. As a result, the number of brain-dead pancreas transplants also increased, reaching 49 cases in 2019. The rate of pancreas transplants per brain-dead donor was 100% in 2009; however, as brain dead donation increased, the rate gradually fell to 50% in 2018 and 2019 [28]. With this decrease in the utilization rate of pancreatic grafts for organ transplantation, pancreatic grafts from brain-dead donors began to be used for islet transplantation in 2013 [29]. Despite the higher rate of utilization of pancreatic grafts from expanded criteria donors, at 1, 5, and 10 years after pancreas transplantation, patient survival rates were 95.8%, 94.2%, 88.7%, respectively, pancreas graft survival rates were 85.9%, 76.2%, and 67.4%, and kidney survival rates were 93.2%, 90.8%, and 78.2% 28, which were comparable to rates in other countries. However, looking at pancreatic graft survival by category, the 1-, 5-, and 10-year pancreatic graft survival rates were 87.3%, 83.2%, and 74.6%, respectively for the simultaneous pancreas and kidney transplantation and 85.4%, 52.3%, and 41.8% for pancreas transplantation after kidney transplantation. In the case of pancreas transplantation, the 1-year graft survival rate was 66.7%, and the 5-year graft survival rate was 31.2%; no data were available for the 10-year graft survival rate. With the categories of pancreas transplantation after kidney and pancreas transplantation alone being associated with worse graft survival in Japan, islet transplantation has become a promising treatment for patients with type 1 diabetes.

## 3. The Dawn of Clinical Islet Transplantation in Japan; 1997–2002

In 1996, volunteers from Chiba University, Tokyo Women’s Medical University, University of Tsukuba, and National Sakura Hospital gathered held a meeting in Tokyo to start activities for clinic islet transplantation. This meeting officially became the Working Group of Islet Transplantation in the Japanese Pancreas and Islet Transplantation Association (JPITA) in 1997 and has continued to this day (Figure 1).

To date, the Working Group of Islet Transplantation has conducted feasibility studies for the implementation of clinical islet transplantation in Japan and has defined and standardized the donor criteria for islet transplantation, the recipient criteria, and the facility criteria for islet isolation/transplantation. This working group also published several documents for the purpose of the achievement of safe and effective clinical islet transplantation, including guidelines and manuals [30]. These documents were eventually summarized in the Manual for Clinical Islet Transplantation in Japan, 4th edition, which was published at the time when islet transplantation received national health insurance coverage in 2020 [31].

Since then, the secretariat of islet transplants in this working group has carried out the certification work for islet transplant facilities, recipient registration, and coordination and allocation of pancreatic grafts when receiving donor information. However, as mentioned in the previous chapter, even the number of cases of organ transplantation was quite small in Japan during this era. This was due to the law, the system based on it, and Japanese views and thoughts in relation to life and death. It was necessary to build a system in which the clinical staff engaged in islet transplantation can obtain donor information even for potential donors. This required obtaining a certain degree of national consensus. Thus, it took a long time for clinical islet transplantation to start. After spending approximately five years on such preparations, clinical islet transplantation has finally begun in Japan.

By adding some topics of this era, some translational research was subsequently applied to clinical islet transplantation in Japan and in some other countries. Kuroda et al. [32,33] reported that pancreatic preservation by the two-layer method improved the islet yield, and this method was used clinically in various countries, including Japan. Kenmochi et al. [34] created a mechanical chopping machine for the first phase of digestion of pancreata and used it for subsequent clinical islet isolation in Japan. More than 20 years of history of clinical islet transplantation in Japan, from the first official meeting until clinical islet transplantation, received national health insurance coverage.

## 4. Clinical Islet Transplantation in the Edmonton Protocol Era until Interruption Due to the Possible Contamination of Collagenase by Bovine Brain Component; 2003–2007

The criteria for islet transplantation during this period are as follows: (1) Intrinsic insulin secretion is significantly reduced, and insulin treatment is required. (2) More than five years have passed since the onset of type 1 diabetes. (3) Diabetes is difficult to control, even by a physician who is a specialist in the treatment of diabetes. (4) In principle, patients under 75 years of age. From the viewpoint of the renal function, diabetic nephropathy up to stage 3 is included in the case of islet transplantation alone, and in the case of islet transplantation after kidney transplantation, the serum creatinine level should be ≤1.8 mg/dL at ≥6 months after kidney transplantation, and patients in whom the serum creatinine level is elevated by ≥0.2 mg/dL and/or who require an oral steroid dose of ≥10 mg/dL to maintain their renal function were excluded.

In 2003, the Chiba East Hospital group performed the first islet isolation from a donor after cardiac arrest in Japan [35]. However, the islet yield at this time was only 177,800 IEQ, which did not meet the transplantation criteria of 5000 IEQ per recipient weight (kg), and islet transplantation was not performed. The first case of islet transplantation in Japan was performed in 2004 by the Kyoto University group, and since then, 65 islet isolations have been performed, while islet transplantation has been performed 34 times in 18 cases (5 males and 13 females) from 2004 to 2007 [36,37]. In this era, most pancreatic grafts for islet transplantation were performed with donation after cardiac death (DCD), without the withdrawal of life-sustaining therapies. Although the pancreas graft donation from brain-dead donors for islet transplantation was not prohibited, as mentioned in Chapter 2, one of the reasons for the lack of donation was that there was an overwhelming shortage of brain-dead donors in Japan in this era. As noted above, there was no system for islet transplant centers to obtain information on brain-dead donors. As noted later, islet transplantation from brain-dead donors has been carried out since 2013, and the success rate of islet isolation has improved. Although these Japanese donors seem to fit category 4 according to the Modified Maastricht classification [38], DCD in Japan does not fit anywhere in the Modified Maastricht classification because most DCD in Japan is not associated with the withdrawal of life-sustaining therapies, such as turning a respirator off.

These clinical islet transplants were performed according to the Edmonton Protocol. With the establishment of an islet isolation method that yielded a sufficient amount of high-quality pancreatic islets, this protocol eliminates steroids, which were conventionally used as immunosuppressants. An anti-IL-2 receptor monoclonal antibody was used for induction immunosuppressive therapy, and sirolimus and small dose tacrolimus were also used as maintenance immunosuppressive therapy. Pancreatic islets obtained from multiple donors (up to three) should be transplanted via the portal vein into one recipient in a relatively short period of time for maximum efficacy (e.g., achievement of insulin independence) [7]. Although the Edmonton Protocol envisions multiple islet transplantations from a maximum of three donors per recipient, 18 cases of islet transplantation were performed in Japan in this era [36,37], among which islet transplantation was performed once in eight cases, twice in four cases, and three times in six cases because of a shortage of donors. These results of islet transplantation performed in this era in Japan demonstrated safety and minimal invasiveness. However, among these cases, insulin independence was temporarily achieved in only three cases (one case with two transplants and two cases with three transplants); however, the maximum duration of insulin independence was 214 days. The 1-, 2-, and 5-year islet graft survival rates (defined by a C-peptide level of ≥0.3 ng/mL) after the first transplant were 72.2%, 44.4%, and 22.2%, respectively. Even if you pay attention to the six recipients who received three transplants, the outcomes of islet transplantation in Japan have not been good in this era [36].

In this era, due to the absolute shortage of donors, the Kyoto University group also attempted to perform islet transplantation using a partial pancreatic graft from a living donor [39,40,41], but it has not been continued since then.

In 2007, it was discovered that bovine brain component was used in the manufacturing process of collagenase liberase for islet isolation, and the Ministry of Health, Labor, and Welfare in Japan issued a recommendation to discontinue islet isolation for clinical transplantation using liberase due to the possible risk of infection with transmissible spongiform encephalopathy (TSE), specifically mad cow disease. Since then, clinical islet transplantation in Japan has been interrupted. According to a Japanese survey, no cases of TSE associated with islet isolation using liberase were observed in Japan [42]. At that time, the use of enzymes other than Liberase for islet isolation (e.g., Serva enzyme) was also considered in Japan. However, clinical islet transplantation was performed as part of a single-center study at that time. It was not possible to carry out clinical islet transplantation using these, which was due to their higher cost.

## 5. Multicenter Clinical Islet Transplantation Study and Utilization of Pancreatic Grafts from Brain-Dead Donors for Islet Transplantation; 2012–2019

In 2012, “Islet transplantation using brain-dead donors and donors after cardiac death for patients with insulin-dependent diabetes mellitus suffering from complicating hypoglycemia unawareness” (CIT-J003) was started as a multicenter clinical trial with the aim of gaining national health insurance coverage. In addition, clinical islet transplantation in Japan was carried out in the category of tissue transplantation in compliance with the guidelines of the JSTT, However, in 2014, the law on ensuring the safety of regenerative medicine (details will be described later) was enforced, and it was decided that clinical islet transplantation would be included in that category. It was reported that the long-term outcomes of islet transplantation were improved by induction therapy with anti-thymocyte globulin and anti-TNFα antibody followed by maintenance therapy with low-dose tacrolimus, sirolimus, or mycophenolate mofetil in a multicenter phase III clinical trial [12]. By following this protocol, a multicenter phase II clinical trial was conducted in Japan in order to verify the safety and usefulness of this protocol, which involves the administration of anti-thymocyte globulin—a T cell depleting antibody—at the time of the induction of immunosuppression, and the use of a calcineurin inhibitor and mycophenolate mofetil for maintenance immunosuppression. The primary endpoint of this study was the proportion of patients with HbA1c < 7.4% without severe hypoglycemic attacks from 90 days to 365 days after the first islet transplantation.

Nine patients received their first islet transplantation, five received their second transplantation, and two received their third transplantation. Eight of these cases were included in the efficacy analysis population. The interim analysis of this study revealed excellent results, with the primary endpoint achieved in 75% of cases, and islet graft survival (defined as a C-peptide level of 0.3 ng/mL or higher) at 4 years after the first islet infusion achieved in 80% of the cases indicated with the Kaplan–Meier curve [43]. In addition, there were two cases that met the criteria for insulin independence. The results of these interim analyses fulfilled the criteria for early discontinuation (effective discontinuation) of the study and demonstrated the safety and lower invasiveness of this protocol. The study was completed in October 2019.

Initially, in this study, islet transplantation was only performed using pancreatic grafts from donors after cardiac arrest; however, with the increase in brain-dead donors, islet transplantation using pancreases from brain-dead donors was started in 2013. This should be considered to have significantly contributed to the improvement of the outcomes of this study. While the median islet yield was 275,550 IEQ using pancreatic grafts from donors after cardiac arrest, the median islet yield from brain dead donors was 362,700 IEQ, which represents a significant improvement. While successful islet isolation (>5000 IEQ per recipient weight) was achieved in 38 of 71 cases (53.5%) in which pancreatic grafts from donors after cardiac arrest were used, it was achieved in 12 of 15 cases (80.0%) in which brain-dead donors were used [29].

## 6. Islet Transplantation under National Health Insurance; 2020–Present

Following the satisfactory results of a multicenter clinical trial for islet transplantation, allogeneic islet transplantation received national health insurance coverage from April 2020. At the same time, the standards for facilities that perform islet transplantation were set. The outline of medical fees and facility standards are as follows:Overview of medical fees under national health insurance.

In the case of islet transplantation covered by national health insurance, the patient would owe approximately 30% of the medical costs. However, if the medical expenses exceed a certain amount (depending on their income), there is a high medical expenses system, which exempts many patients from paying medical expenses exceeding 80,000 yen per month. Since all clinical islet transplants in Japan before the introduction of medical insurance were clinical trials, medical personnel secured a huge amount of research funds and spent them on the islet transplants. With the introduction of insurance coverage for medical treatment, islet transplantation has come to be performed as general medical treatment (without obtaining research funds);


2.The facility Standards for clinical allogeneic islet transplantation under national health insurance coverage are as follows:
(1)Five or more cases of allogeneic pancreas transplantation and/or islet transplantation have been performed in 3 years;(2)Two or more full-time surgeons, including one or more with experience of at least 3 cases of islet transplantation;(3)Two or more full-time physicians who are specialists in diabetes and have experience of 5 years or more in the treatment of diabetes, with one or more of these physicians having experience in treating patients with pancreas transplantation or islet transplantation;(4)The facility must be accredited by JPITA for the performance of islet transplantation;(5)Compliance with “Guidelines on Ethical Issues in Application of Human Tissue to Medical Practice” and “Guidelines on the Safety, Storage, and Application of Human Tissue in Medical Practice” defined by the Japanese Society for Tissue Transplantation;(6)Compliance with the standards for providing regenerative medicine as stipulated in Article 3 of the Act on Ensuring the Safety of Regenerative Medicine.


“The Act on Ensuring Safety of Regenerative Medicine (Regenerative Medicine Safety Assurance Act)” has been enforced since 25 November 2014, in Japan, in order to clarify the measures used by people who implement regenerative medicine. Under the Regenerative Medicine Safety Assurance Law, medical procedures are classified into three categories (Categories 1, 2, and 3) according to the degree of impact on human life and health. Islet transplantation is the only tissue transplant procedure that is subject to this law, and clinical islet transplantation is classified as “categories 1 regenerative medicine”, which requires the most legitimate and strict procedures.

In carrying out islet transplantation, it is also necessary to comply with the “Guidelines on the Safety, Storage, and Application of Human Tissue in Medical Practice” established by the JSTT. Table 1 shows the donor criteria for islet transplantation shown in these guidelines. These donor criteria in relation to infectious diseases of the donor for tissue transplantation (islet transplantation is categorized as tissue transplantation in Japan) are described in great detail. In addition, it is noteworthy that donation following death after abuse (in patients of <18 years of age) is defined as an exclusion criterion.

Diabetes, pancreatitis, and alcoholism are mentioned as exclusion criteria peculiar to islet transplantation; however, no specific numerical values (e.g., HbA1c) are defined. In a previous multicenter clinical trial, HbA1c ≥ 6.0% was set as a criterion for exclusion of donors; however, since the HbA1c value of the donor is sometimes affected by anemia, each islet transplant center can evaluate the glucose tolerance of the donor candidate considering the time course of blood glucose, including insulin requirement in the intensive care unit, as well as HbA1c.

The recipient criteria for islet transplantation are shown in Table 2. The committee for judgment of the indications for islet transplantation in Japanese patients, The Pancreas and Islet Transplantation Association, judges the compatibility with these indication criteria for each patient. The patients for whom islet transplantation is found to be indicated are registered on the waiting list. Briefly, the recipient candidate should be 20 to 75 years of age. The recipient candidates also suffered from diabetes with insulin dependence that had persisted for more than 5 years, showed a severe decrease in endogenous insulin secretion (serum C-peptide < 0.2 ng/mL), and had significant difficulty in blood glucose control, even when the patient’s treatment was managed by a diabetes specialist. The cases where blood glucose control is difficult due to anti-insulin antibodies or autonomic neuropathy should also be involved as the candidate for islet transplantation is judged by the committee for judgment of indications for islet transplantation.

At the time of donation for islet transplantation, the rules for recipient selection differed between donation after cardiac death and brain dead donation. The recipient selection criteria for islet transplantation in each case are shown in Table 3. It is noteworthy that regionality is the first condition for recipient selection in cases of islet transplantation from donors after cardiac death in Japan because most pancreatic graft recovery teams must wait for a long time without withdrawing life-sustaining therapies, such as respirator-off. Under national health insurance coverage, the criteria for the transplantation of isolated islets depend on each islet transplant center. The criteria of most of the centers are similar to those that were used in a multicenter clinical trial, with minor modification, as indicated in Table 4.

## 7. Concluding Remarks

In April 2020, clinical islet transplantation marked a major turning point by receiving national health insurance coverage in Japan. The outcomes of islet transplantation under national health insurance coverage are yet to be reported. The fact that this minimally invasive treatment has now received health insurance coverage will be very beneficial to patients with type 1 diabetes who have severely unstable glycemic control. However, due to the COVID-19 pandemic, both brain-dead and cardiac arrest donors have been decreasing since 2020, both in Japan and elsewhere in the world [44]. The further development of all types of transplantation requires an increase in organ donations, and it is considered that the change in the number of organ donations after the COVID-19 pandemic will greatly affect the success or failure of islet transplantation in the future

## Figures and Tables

**Figure 1 jcm-11-01645-f001:**
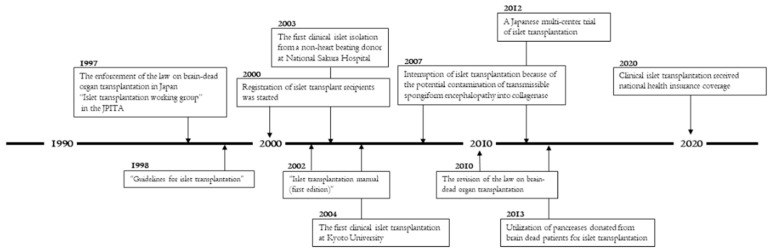
Remarkable events for islet transplantation in Japan.

**Table 1 jcm-11-01645-t001:** Donor criteria for islet transplantation.

① As a general rule, the age should be 70 years or younger. ② The warm ischemic time (time from cardiac arrest to the start of perfusion into pancreatic graft in donation after cardiac death) should be within 30 min. ③ Exclusion criteria for tissue transplantation including islet transplantation (1) Systemic active infections (bacteria, fungi, viruses, etc.) (Note 1) Local infectious diseases, such as pneumonia, are judged based on the judgment of the collection team and the test results after collection. (2) Positive for syphilis, HBs antigen, HCV antibody (Note 2), HTLV-1 antibody, or HIV antibody (3) Creutzfeldt–Jakob disease or suspected Creutzfeldt–Jakob disease (4) Malignant disease (Note 3), including hematopoietic tumors, such as leukemia and malignant lymphoma (5) Serious metabolic/endocrine diseases, autoimmune diseases, such as blood diseases and collagen diseases (6) Death of unknown cause (7) Death after abuse (under 18 years of age) Note 1: The main target viral infections are as follows Parvovirus B19 infection, West Nile virus infection, SARS (severe acute respiratory syndrome) infection, rabies, amoeba encephalitis due to Balamcia mandrills infection, Zika virus disease Note 2: Donation should be possible in cases of complete remission with SVR and negativity for HCV-RNA Note 3: Exclude basal cell carcinoma of the skin and primary brain tumors. In addition, exclude complete remission diagnosed by the doctor in charge of tumor treatment. ④ Exclusion criteria specific for islet transplantation (1) Diabetes (2) Acute/chronic pancreatitis (3) Alcoholism (4) Not suitable for transplantation due to functional or organic damage to the pancreas

**Table 2 jcm-11-01645-t002:** Indication criteria for islet transplantation.

**Patients who should be considered for islet transplantation** Patients with abolished endogenous insulin secretion who suffer from blood glucose instability, including severe hypoglycemia (hypoglycemia requiring the assistance of a third party), even with specialized treatment. **Indication criteria** 1. Consent of the person regarding islet transplantation 2. Age 20 to 75 years (at the time of consent) 3. Insulin dependence persists for more than 5 years 4. Severe decrease in endogenous insulin secretion (serum C-peptide < 0.2 ng/mL) 5. Significant difficulty in achieving blood glucose control, even with treatment efforts by a diabetes specialist 6. Approval by the committee for the judgment of indications for islet transplantation in a case in which blood glucose control is difficult due to anti-insulin antibodies or autonomic neuropathy **Exclusion criteria** 1. Moderate or higher obesity (BMI ≥ 30) 2. Severe ischemic heart disease or heart failure 3. Severe liver dysfunction 4. Severe renal dysfunction (eGFR ≤ 30 mL/min/1.73 m^2^. However, patients who have undergone renal transplantation should be evaluated individually.) 5. Unstable retinopathy (excluding blindness) 6. Alcohol dependence or drug dependence 7. Active and latent infections that may be exacerbated under post-transplant immunosuppression 8. Active foot ulcer/gangrenous lesion 9. Malignant tumor 10. Other factors that are considered to make the case unsuitable for transplantation

The Japanese Pancreas and Islet Transplantation Association, The committee for the judgment of indications for islet transplantation 2020.3.

**Table 3 jcm-11-01645-t003:** Recipient selection criteria for islet transplantation.

Recipients will be selected preferentially in the following order. **In cases of donation after cardiac death** (1) Regionality (2) ABO blood type identical (3) Second-transplantation, 3rd-transplantation * (However, there is no priority between 2nd-transplantation and 3rd-transplantation) (4) Waiting period (calculated from the last transplant date for 2nd-transplant and 3rd-transplant) If there is no blood type matched candidate, the selection order is determined from blood type mismatched candidates. Perform lymphocyte cross-matching before transplantation. At least, a negative result for direct cross-match of warm T cell is needed. * Second-transplantation and 3rd-transplantation should be cases in which islet transplantation has already been performed, and the graft function has been confirmed, but insulin withdrawal has not been obtained. Cases in which islet transplantation has already been performed and in which the graft function cannot be confirmed are not prioritized for 2nd-transplantation or 3rd-transplantation. **In cases of brain dead donation** (1) ABO blood type identical (2) (2) 2nd-transplantation, 3rd-transplantation * (However, there is no priority between 2nd-transplantation and 3rd-transplantation) (3) Waiting period (calculated from the last transplant date for 2nd-transplant and 3rd-transplant) (4) Regionality If there is no blood type matched candidate, the selection order is determined from blood type mismatched candidates. Perform lymphocyte cross-matching before transplantation. At least, a negative result for direct cross-match of warm T cell is needed. * 2nd-transplantation and 3rd-transplantation should be the cases in which islet transplantation has already been performed, and the graft function has been confirmed, but insulin withdrawal has not been obtained. Cases that have already undergone islet transplantation and whose graft function cannot be confirmed are not prioritized for 2nd-transplantation or 3rd-transplantation.

**Table 4 jcm-11-01645-t004:** Islet isolation criteria for islet transplantation.

Islets transplantation can be performed when the isolated islets meet the following conditions. Islet yield ≥ 5000 IE/kg (patient weight) Purity ≥ 30% Tissue volume ≤ 10 mL Viability ≥ 70% Endotoxin ≤ 5EU/kg (patient weight) Gram stain negative

## Data Availability

Not applicable.

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
