# Peer review of "The History of Clinical Islet Transplantation in Japan"

_jcm, 2022, doi:10.3390/jcm11061645_

Round 1
Reviewer 1 Report
I have read with great interest the manuscript entitled, “The history of clinical islet transplantation in Japan” by Ito T and co-workers. This is a comprehensive review manuscript covering the activity and progress of islet transplantation in Japan over 20 years. The manuscript has a significant potential and is valuable for readers not only in Japan but also in other countries. However, there are some major points as well as minor points which the authors should address in order to improve the manuscript.
Section 2. Pancreas transplantation in Japan
[1] Regarding the amended law, the authors should describe details on what the exact changes in the amended law were and what the issues with the previous law were.
[2] Clinical outcomes of pancreas graft in PTA and PAK are not as good as those in SPK. This finding is well recognized worldwide, and is not a specific finding in Japan. Therefore, it is not logical for the authors to justify the use of brain-dead donors in islet transplant based on the inferior outcomes of PTA and PAK.
Section 3. The dawn of clinical islet transplantation in Japan; 1997-2002
[3] In this section, the authors describe many non-clinical activities. This is fine, but listing titles of documentations, which are not accessible to the readers, is not helpful at all. “Guidelines for islet transplantation”, “Do you know islet transplantation?”, “Islet transplantation”, “Donation of pancreatic graft for islet transplantation”, “Outline of islet transplantation”, “Agreement for islet transplantation”, and “Manual for clinical islet transplantation in Japan”: all should be removed from the text as well as from the Figure 1. Moreover, the authors should describe the findings from the feasibility studies conducted during this era.
[4] The authors should describe and emphasize the followings in this section:
- The reasons why clinical islet transplantation was not performed at all.
- Transplant surgeons in Japan were trying to set rules for clinical practice on a national level, not an individual institutional level.
- Many other countries started islet transplantation programs after the publication of the Edmonton protocol. In contrast, transplant community in Japan was trying to start islet transplantation prior to the Edmonton protocol. This should be emphasized.
Section 4. Clinical islet transplantation in the Edmonton protocol era until interruption due to the possible contamination of collagenase by bovine serum; 2003-2007
[5] First of all, the authors should edit the title of this section. Bovine brain component, not bovine serum, was an issue. Accordingly, “…bovine serum…” on page 7, line 268 and 280 as well as in the abstract should be edited.
[6] The authors state that most pancreas grafts for islet were donated from DCD in this era. The authors should explain why DCD, not brain-dead donors, were used in this era. Was using pancreases from brain-dead donors prohibited?
[7] Many centers in the world continued islet transplants without interruption, or restarted the activity after a short period of interruption, utilizing an alternative enzyme product. The authors should explain to the readers why there were no activities in Japan until 2012 and why an alternative enzyme product was not used. Also, the authors may want to describe what was done within the islet transplant community in Japan between 2008 and 2011.
[8] On page 7, line 285: “According to … have been observed”: This sentence is confusing and difficult to understand. Were there cases of TSE associated with Liberase in Japan? This is a critical point.
Section 5. Multicenter clinical islet transplantation study and utilization of pancreatic grafts from brain-dead donors for islet transplantation; 2012-2019
[9] “Islet graft survival at 4 years after the first islet infusion achieved in 80% of the cases”. This means 6.4 patients achieved graft survival at 4 years (80% of 8 patients gives 6.4 patients). Is 80% coming from the Kalan-Meier estimate? If so, please clarify and modify the sentence.
[10] The authors’ description about uncontrolled DCD and Modified Maastricht classification is simply wrong. Uncontrolled DCD refers to circumstances where donation is considered after death has occurred without anticipation. In contrast, controlled DCD refers to circumstances where donation may initially be considered when death is anticipated, but has not yet occurred. “Controlled” and “Uncontrolled” are not associated with timing of death. Even after withdrawal of life supporting therapy, no one can “control” timing of death. It takes less than 30 minutes in some cases and it may take more than 1 hour in other cases from withdrawal of life supporting therapy to death. Many Japanese transplant surgeons seem to misunderstand what uncontrolled DCD refers to. Finally, DCD in Japan does not fit anywhere in the Modified Maastricht classification.
- Islet transplantation under national health insurance; 2020-present
[11] The authors describe medical fees under subsection 1. The readers do not understand what these descriptions of medical fees mean. Listing actual fees is meaningless without knowing the real health care system in Japan because the health care insurance system varies from country to country. Please describe, both from the patient perspective and from the health care professional perspective, what will be changed after implementing the health care coverage from the government.
[12] It is unclear if there are any differences between category 1 and type 1 regenerative medicine.
[13] According to the authors’ statement, islet transplantation is classified as type 1 because it involves the transplantation of cells from another person. This indicates that transplantation of other cells such as bone marrow cells is also classified as type 1. Please clarify.
[14] A paragraph starting from “In carrying out…” simply duplicates information contained in Table 1. Rather than listing the donor criteria in the text, please describe the unique points in Japan comparing to other countries.
[15] What is missing here in this section is that the release criteria of isolated islets for transplantation.
[16] Finally, references 19, 20, 31, and 32 should be removed as these are not necessary and not helpful to the readers.
Minor points
English syntax is incorrect in many places throughout the text and several sentences are difficult to understand as written.
[17] “Type 1 diabetic patients” (in the abstract) should be “patients with type 1 diabetes”. The same applies to “diabetic patients” in Table 2.
[18] On page 1, line 43: “Therefore, islet…” A conjunctive adverb “therefore” does not fit here. There are many misusages of conjunctive adverbs throughout the text.
[19] On page 2. line 46: “..and the pancreas grafts have to be shared with patients waiting for…”. This is not appropriate expression. Organs or grafts cannot be shared with persons.
[20] On page 3, line 109: “pancreas translation” should be “pancreas transplantation”.
[21] On page 3, line 130: “31” should be “[31]”. There are many other similar errors throughout the text.
[22] Table 1: “Creutzfledt-Jakob disease suspected Creutzfledt-Jakob disease” should be “Creutzfledt-Jakob disease or suspected Creutzfledt-Jakob disease”.
[23] Table 3: Avoid using “matching, “matched”, and “mismatched” for ABO blood typing. Use “identical” and “compatible”.
[24] Table 3: “Perform lymphocyte cross matching before transplantation”. Then what? The result has to be negative? Or result does not matter? Needs clarification.
[25] Table 3: “No priority for 2nd-transpantation or 3rd-transplantation”. Do authors mean “no priority between 2nd-transpantation and 3rd-transplantation”?
[26] On page 7, line 263: “Although the Edmonton…by serum bovine”: it is too long and hard to understand what the authors are trying to say. Try not saying about collagenase issue here because it is mentioned in the last paragraph.
[27] On page 7, line 272: “Even when…invasiveness of islet transplantation”: the sentence structure is difficult to read, and it is hard to understand what the authors are trying to say.
[28] On page 7, line 305-309: Two identical sentences.
[29] On page 8, line 332: “…facility standards for facilities…”: This is unclear.
[30] On page 8, line 350: Please spell out JPITA.
[31] On page 9, line 368: Please spell out JSTT.
[32] On page 9, line 380: “The committee…list.”: Better to split this sentence into two with modification.
Author Response
We wish to express our strong appreciation to both reviewers for their insightful comments on our paper. We believe that the comments have helped us to significantly improve the paper.
RESPONSE TO REVIEWER 1
Major points;
Comment 1
Regarding the amended law, the authors should describe details on what the exact changes in the amended law were and what the issues with the previous law were.
Response:
The following text has been added (P.2, L.96);
“Before the amendment of the law, brain death was limited to human death only in the case of donation for organ transplantation and since brain-dead organ donation required the person to make their intention clear before brain death, for the first 10 years (approximately) brain-dead organ transplantation was only performed in a few cases each year. The primary modification of the law was that declaration of a person's intent to donate before brain death is no longer required for brain-dead donation. ”
Comment 2
Clinical outcomes of pancreas graft in PTA and PAK are not as good as those in SPK. This finding is well recognized worldwide, and is not a specific finding in Japan. Therefore, it is not logical for the authors to justify the use of brain-dead donors in islet transplant based on the inferior outcomes of PTA and PAK.
Response:
We never stated that the poor results of PTA and PAK are the only reasons for starting the use of pancreata from brain-dead donors in islet transplants. The main reason is that as brain dead donation increased, the usage of the pancreas for organ transplantation gradually fell. We analyzed whether pancreata that were not used for organ transplantation could be used for islet transplantation. The details are described in Reference 29.
Comment 3
In this section, the authors describe many non-clinical activities. This is fine, but listing titles of documentations, which are not accessible to the readers, is not helpful at all. “Guidelines for islet transplantation”, “Do you know islet transplantation?”, “Islet transplantation”, “Donation of pancreatic graft for islet transplantation”, “Outline of islet transplantation”, “Agreement for islet transplantation”, and “Manual for clinical islet transplantation in Japan”: all should be removed from the text as well as from the Figure 1. Moreover, the authors should describe the findings from the feasibility studies conducted during this era.
Response:
"Do you know islet transplantation?" which was for patients has been deleted. However, we consider that listing the titles of other documents is could be helpful. We think that the fact that that the fact that these documents were needed to achieve the clinical introduction of islet transplantation in Japan is useful information for the readers. These documents, including the guidelines, were eventually summarized in the Manual for clinical islet transplantation in Japan, 4th edition. The Manual for clinical islet transplantation in Japan, 4th edition (both Japanese and English versions) will be published on the homepage of The Japanese Pancreas and Islet Transplantation Association and will be widely accessible to readers.
The text has been changed below (P.3, L.137);
“These documents, including the guidelines, were eventually summarized in the Manual for clinical islet transplantation in Japan, 4th edition, which was published at the time that islet transplantation received national health insurance coverage in 2020 [32].”
The following text has been added (P.4, L.152);
“Adding as some topics of this era, some translational research was subsequently applied to clinical islet transplantation in Japan and in some other countries. Kuroda et al. [33,34] reported that pancreatic preservation by the two-layer method improved the islet yield, and this method was used clinically in various countries, including Japan. Iwashita et al. [35] created a mechanical chopping machine for the first phase of digestion of pancreata and used it for subsequent clinical islet isolation in Japan.”
Comment 4
The authors should describe and emphasize the followings in this section:
- The reasons why clinical islet transplantation was not performed at all.
- Transplant surgeons in Japan were trying to set rules for clinical practice on a national level, not an individual institutional level.
- Many other countries started islet transplantation programs after the publication of the Edmonton protocol. In contrast, transplant community in Japan was trying to start islet transplantation prior to the Edmonton protocol. This should be emphasized.
Response:
- The reasons why clinical islet transplantation was not performed at all.
As mentioned in the previous chapter, even the number of cases of organ transplantation was quite small in Japan at this era. This was due to the law, the system based on it, and Japanese views and thoughts in relation to life and death. It was necessary to build a system in which the clinical staff engaged in islet transplantation can obtain donor information even for potential donors. This required obtaining a certain degree of national consensus. Thus, it took a long time for clinical islet transplantation to start.
The following text has been added (P.3, L.143);
“However, as mentioned in the previous chapter, even the number of cases of organ transplantation was quite small in Japan during this era. This was due to the law, the system based on it, and Japanese views and thoughts in relation to life and death. It was necessary to build a system in which the clinical staff engaged in islet transplantation can obtain donor information even for potential donors. This required obtaining a certain degree of national consensus. Thus, it took a long time for clinical islet transplantation to start.” 
- Transplant surgeons in Japan were trying to set rules for clinical practice on a national level, not an individual institutional level.
- Many other countries started islet transplantation programs after the publication of the Edmonton protocol. In contrast, transplant community in Japan was trying to start islet transplantation prior to the Edmonton protocol. This should be emphasized.
As the reviewer pointed out, before the release of the Edmonton Protocol, each country had no choice but to perform experimental clinical islet transplantation using their own protocol. One of the preparations for clinical practice on a national level in Japan was the publication of the numerous above-mentioned documents. Other things that were required to start islet transplantation included recipient registration, and coordination and allocation of pancreatic grafts. These details are mentioned in Section 3.
Comment 5
First of all, the authors should edit the title of this section. Bovine brain component, not bovine serum, was an issue. Accordingly, “…bovine serum…” on page 7, line 268 and 280 as well as in the abstract should be edited.
Response:
The word "bovine serum" was changed to "Bovine brain component".
Comment 6
The authors state that most pancreas grafts for islet were donated from DCD in this era. The authors should explain w
hy DCD, not brain-dead donors, were used in this era. Was using pancreases from brain-dead donors prohibited?
Response:
Donation of pancreas grafts from brain-dead donors for islet transplantation was not prohibited. As mentioned in Section 2, one of the reasons was that there was an overwhelming shortage of brain-dead donors. At one point, 100% of brain-dead donors were used for pancreas transplantation, which I think is unprecedented in the world. There was no system for the islet transplant centers to obtain information on brain-dead donors. Brain-dead donor information was systematized under the legislation, but islet transplantation was not included in this category in Japan. Even now, this is not managed under the legal system; however, the Japanese Ministry of Health, Labor and Welfare has requested at islet transplant centers be provided with as much brain-dead donor information as possible, and it has become possible to obtain brain-dead donor information.
The manuscript was modified as follows (P.7, L.283);
“In this era, most pancreatic grafts for islet transplantation were performed with donation after cardiac death (DCD), without the withdrawal of life-sustaining therapies. Although the pancreas graft donation from brain-dead donors for islet transplantation was not prohibited, as mentioned in Chapter 2, one of the reasons for the lack of donation was that there was an overwhelming shortage of brain-dead donors in Japan in this era. As noted above, there was no system for islet transplant centers to obtain information on brain-dead donors. As noted later, islet transplantation from brain-dead donors has been carried out since 2013, and the success rate of islet isolation has improved.”
Comment 7
Many centers in the world continued islet transplants without interruption, or restarted the activity after a short period of interruption, utilizing an alternative enzyme product. The authors should explain to the readers why there were no activities in Japan until 2012 and why an alternative enzyme product was not used. Also, the authors may want to describe what was done within the islet transplant community in Japan between 2008 and 2011.
Response:
First, islet transplantation was discontinued at the request by the Ministry of Health, Labor, and Welfare in Japan. They recommend using mammalian free enzyme for islet transplantation.
Mammalian free enzyme (Liberase MTF-S) was introduced to the Japanese market later than in other countries. We also hoped that the resumption of clinical islet transplantation would be analyzed in a multi-center clinical trial that would lead to national health insurance coverage. The year 2012 was the optimal time for using mammalian free enzyme and for starting the new multi-center clinical trial.
Comment 8
On page 7, line 285: “According to … have been observed”: This sentence is confusing and difficult to understand. Were there cases of TSE associated with Liberase in Japan? This is a critical point.
Response:
This was an error. This sentence has been corrected as follows (P.7, L.323);
“According to a Japanese survey, no cases of TSE associated with islet isolation using Liberase have been observed in Japan”
Comment 9
“Islet graft survival at 4 years after the first islet infusion achieved in 80% of the cases”. This means 6.4 patients achieved graft survival at 4 years (80% of 8 patients gives 6.4 patients). Is 80% coming from the Kalan-Meier estimate? If so, please clarify and modify the sentence.
Response:
As the reviewer pointed out, this analysis was performed using Kaplan-Meier curves. At 4 years after transplantation, 5 cases could be observed, 4 of which were found to be positive for C-peptide.
The following phrase has been added (P.8, L.350);
“indicated with the Kaplan-Meier curve”
Comment 10
The authors’ description about uncontrolled DCD and Modified Maastricht classification is simply wrong. Uncontrolled DCD refers to circumstances where donation is considered after death has occurred without anticipation. In contrast, controlled DCD refers to circumstances where donation may initially be considered when death is anticipated, but has not yet occurred. “Controlled” and “Uncontrolled” are not associated with timing of death. Even after withdrawal of life supporting therapy, no one can “control” timing of death. It takes less than 30 minutes in some cases and it may take more than 1 hour in other cases from withdrawal of life supporting therapy to death. Many Japanese transplant surgeons seem to misunderstand what uncontrolled DCD refers to. Finally, DCD in Japan does not fit anywhere in the Modified Maastricht classification.
Response:
As the reviewer pointed out, I totally agree that "Japanese DCD does not fit anywhere in the modified Maastricht classification."
However, in reference 36, the author noted that "In some countries, where the legislation does not accept brain death criteria (i.e. Japan) or when the patient will never meet the neurological criteria for the diagnosis of BD, the procedure for potential DBD can be converted to DCD (controlled CA)".
I mentioned this because it seemed as if donation after cardiac arrest was controlled. Again, I agree that "Japanese DCD does not fit anywhere in the modified Maastricht classification."
The explanation was changed as follows (P.7, L.291);
“Although these Japanese donors seem to fit category 4 according to the Modified Maastricht classification [39], DCD in Japan does not fit anywhere in the Modified Maastricht classification because most DCD in Japan is not associated with the withdrawal of life-sustaining therapies, such as turning a respirator off.”
Comment 11
The authors describe medical fees under subsection 1. The readers do not understand what these descriptions of medical fees mean. Listing actual fees is meaningless without knowing the real health care system in Japan because the health care insurance system varies from country to country. Please describe, both from the patient perspective and from the health care professional perspective, what will be changed after implementing the health care coverage from the government.
Response:
This part was modified following (P.8, L.371);
“In the case of islet transplantation covered by national health insurance, the patient would owe approximately 30% of the medical costs. However, if the medical expenses exceed a certain amount (depending on their income), there is a high medical expenses system, which exempts many patients from paying medical expenses exceeding 80,000 yen per month. Since all clinical islet transplants in Japan before the introduction of medical insurance were clinical trials, medical personnel secured a huge amount of research funds and spent them on the islet transplants. With the introduction of insurance coverage for medical treatment, islet transplantation has come to be performed as general medical treatment (without obtaining research funds).”
Comment 12
It is unclear if there are any differences between category 1 and type 1 regenerative medicine.
Response:
Type 1 was incorrect. Category 1 is correct.
“Type 1” was changed to “Category 1”
Comment 13
According to the authors’ statement, islet transplantation is classified as type 1 because it involves the transplantation of cells from another person. This indicates that transplantation of other cells such as bone marrow cells is also classified as type 1. Please clarify.
Response:
Islet transplantation is covered in the category of law on ensuring the safety of regenerative medicine, however, bone marrow transplantation is not included in this category. Thus, bone marrow transplantation is not included in Category 1 of this law either.
The Japanese Ministry of Health, Labor and Welfare defined the treatments that are covered under the law on ensuring the safety of regenerative medicine, and we cannot explain the medical or scientific reasons. However, bone marrow transplantation was performed as a medical treatment with insurance coverage before the enactment of this new law on ensuring the safety of regenerative medicine, and islet transplantation received insurance coverage after the enactment of this law. We think that there is a difference.
Comment 14
A paragraph starting from “In carrying out…” simply duplicates information contained in Table 1. Rather than listing the donor criteria in the text, please describe the unique points in Japan comparing to other countries.
Response:
This paragraph was changed as follows (P.9, L.409);
“These donor criteria in relation to infectious diseases of the donor for tissue transplantation (islet transplantation is categorized as tissue transplantation in Japan) are described in great detail. In addition, it is noteworthy that donation following death after abuse (in patients of <18 years of age) is defined as an exclusion criterion.
Diabetes, pancreatitis, and alcoholism are mentioned as exclusion criteria peculiar to islet transplantation, however, no specific numerical values (e.g., HbA1c) are defined. In a previous multicenter clinical trial, HbA1c ≥6.0% was set as a criterion for exclusion of donors; however, since the HbA1c value of the donor is sometimes affected by anemia, each islet transplant center can evaluate the glucose tolerance of the donor candidate considering the time course of blood glucose, including insulin requirement in the intensive care unit, as well as HbA1c.”
Comment 15
What is missing here in this section is that the release criteria of isolated islets for transplantation.
Response:
The criteria of isolated islets for transplantation depended on each islet transplant center. The criteria of most of the centers are similar to those that were used in a multi-center clinical trial, with minor modification (see Table 4).
Table 4 was added;
Table 4. Islet isolation criteria for islet transplantation
Islets transplantation can be performed when the isolated islets meet the following conditions.
- Islet yield ≥5,000 IE/kg (patient weight)
- Purity ≥30%
- Tissue volume ≤10 ml
- Viability ≥70%
- Endotoxin ≤5EU/kg (patient weight)
- Gram stain negative
The following text has been added (P.10, L.438);
“Under national health insurance coverage, the criteria for the transplantation of isolated islets depend on each islet transplant center. The criteria of most of the centers are similar to those that were used in a multi-center clinical trial, with minor modification, as indicated in Table 4.”
(P.10, L.444);
“The outcomes of islet transplantation under national health insurance coverage are yet to be reported.”
Comment 16
Finally, references 19, 20, 31, and 32 should be removed as these are not necessary and not helpful to the readers.
Response:
We want to confirm why references 19, 20, 31, and 32 should be removed. Is it because these references were written in Japanese.
Non-English references are used in many reports and references 19 and 20 can be accessed online, while references 31and 32 are published books. Moreover, references 19 and 32 will soon be published online in English.
Because we describe facts that were reported in these references in this review, we do not think that these references should be removed. However, if—as a rule—JCM rejects references that are not written in English, we agree to removing these references.
Minor points
English syntax is incorrect in many places throughout the text and several sentences are difficult to understand as written.
Response:
We are very sorry that this was difficult to understand. This manuscript has been checked by a professional editor who is a native speaker of English.
Comment 17
“Type 1 diabetic patients” (in the abstract) should be “patients with type 1 diabetes”. The same applies to “diabetic patients” in Table 2.
Response:
“Type 1 diabetic patients” was changed to “patients with type 1 diabetes” or just “patients”
Comment 18
On page 1, line 43: “Therefore, islet…” A conjunctive adverb “therefore” does not fit here. There are many misusages of conjunctive adverbs throughout the text.
Response:
This part (P.1, L.43) of sentence was changed to “Therefore, islet transplantation, which is less invasive than pancreas transplantation, should be considered an efficient treatment option for patients with type 1 diabetes”
Comment 19
4 On page 2. line 46: “..and the pancreas grafts have to be shared with patients waiting for…”. This is not appropriate expression. Organs or grafts cannot be shared with persons.
Response:
This sentence was changed as follows (P.1, L.45);
“however, problems still remain, including the cost of islet isolation and post-transplant immunosuppressive therapy, as well as the fact pancreas grafts have to be shared among patients waiting for pancreas transplantation and islet transplantation.”
Comment 20
On page 3, line 109: “pancreas translation” should be “pancreas transplantation”.
Response:
The term “pancreas translation” was changed to “pancreas transplantation”
Comment 21
On page 3, line 130: “31” should be “[31]”. There are many other similar errors throughout the text.
Response:
The reference numbers are enclosed in brackets.
Comment 22
Table 1: “Creutzfledt-Jakob disease suspected Creutzfledt-Jakob disease” should be “Creutzfledt-Jakob disease or suspected Creutzfledt-Jakob disease”.
Response:
“Creutzfledt-Jakob disease suspected Creutzfledt-Jakob disease” in Table 1 was changed to “Creutzfeldt-Jakob disease or suspected Creutzfeldt-Jakob disease”.
Comment 23
Table 3: Avoid using “matching, “matched”, and “mismatched” for ABO blood typing. Use “identical” and “compatible”.
Response:
“ABO blood group matching” in Table 3 was changed to “ABO blood type identical”
Comment 24
Table 3: “Perform lymphocyte cross matching before transplantation”. Then what? The result has to be negative? Or result does not matter? Needs clarification.
Response:
The following sentence was added in Table 3.
“At least, a negative result for direct cross-match of warm T cell is needed.”
Comment 25
Table 3: “No priority for 2nd-transpantation or 3rd-transplantation”. Do authors mean “no priority between 2nd-transpantation and 3rd-transplantation”?
Response:
Yes, as the reviewer pointed out the sentence "there is no priority for 2nd-transplantation or 3rd-transplantation" was changed to “there is no priority between 2nd-transplantation and 3rd-transplantation” in Table 3.
Comment 26
On page 7, line 263: “Although the Edmonton…by serum bovine”: it is too long and hard to understand what the authors are trying to say. Try not saying about collagenase issue here because it is mentioned in the last paragraph.
Response:
The following part of this sentence was deleted.
“and the interruption of clinical islet transplantation due to possible contamination of collagenase by serum bovine”
Comment 27
On page 7, line 272: “Even when…invasiveness of islet transplantation”: the sentence structure is difficult to read, and it is hard to understand what the authors are trying to say.
Response:
This part of sentence was modified following (P.7, L.307);
These results of islet transplantation performed in this era in Japan demonstrated the safety and minimal invasiveness. However, among these cases, insulin independence was temporarily achieved in only 3 cases (1 case with 2 transplants and 2 cases with 3 transplants); however, the maximum duration of insulin independence was 214 days. The 1-, 2-, and 5-year islet graft survival rates (defined by a C-peptide level of ≥0.3 ng/ml) after the first transplant were 72.2%, 44.4%, and 22.2%, respectively. Even if you pay attention to the 6 recipients who received 3 transplants, the outcomes of islet transplantation in Japan have not been good in this era [37].
Comment 28
On page 7, line 305-309: Two identical sentences.
Response:
One of the sentences was deleted.
Comment 29
On page 8, line 332: “…facility standards for facilities…”: This is unclear.
Response:
“facility standards for facilities” was changed to “the standards for facilities”
Comment 30
On page 8, line 350: Please spell out JPITA.
Response:
JPITA is the abbreviation for the Japanese Pancreas and Islet Transplantation Association. The first usage of " Japanese Pancreas and Islet Transplantation Association " is on P3, L.127. We defined the abbreviation on first usage.
Comment 31
On page 9, line 368: Please spell out JSTT.
Response:
JSTT is the abbreviation for Japanese Society of Tissue Transplantation. The first usage of "the Japanese Society of Tissue Transplantation" is on P.2, L.59. We defined the abbreviation on first usage.
Comment 32
On page 9, line 380: “The committee…list.”: Better to split this sentence into two with modification.
Response:
As reviewer pointed out, this part of sentence was modified as follows (P.9, L.420);
“The committee for judgment of the indications for islet transplantation in Japanese patients, The Pancreas and Islet Transplantation Association, judges the compatibility with these indication criteria for each patient. The patients for whom islet transplantation is found to indicated are registered in the waiting list.”

Reviewer 2 Report
This is a very interesting paper, that indeed, as claimed by the authors, could be useful for the implementation of islet transplantation in the world.
It is nicely and clearly written.
May we ask for some minor modifications :
- page 7 line 284 : « According to a survey conducted in Japan, there cases of TSE 285 associated with islet isolation using liberase have been observed [37]. » : how many cases ? three ?
- page 7 line 305 : what was the rationale for choosing 7.4% as a threshold for HbA1c, while other countries have chosen 7% .
- page 8 line 315 : what were the criteria for insulin independence ?
Author Response
We wish to express our strong appreciation to both reviewers for their insightful comments on our paper. We believe that the comments have helped us to significantly improve the paper.
RESPONSE TO REVIEWER 2
Comment 1
page 7 line 284 : « According to a survey conducted in Japan, there cases of TSE associated with islet isolation using liberase have been observed [37]. » : how many cases ? three ?
Response:
This was an error. This sentence has been corrected as follows (P.7, L.323);
“According to a Japanese survey, no cases of TSE associated with islet isolation using Liberase have been observed in Japan”
Comment 2
page 7 line 305 : what was the rationale for choosing 7.4% as a threshold for HbA1c, while other countries have chosen 7% .
Response:
According to the Diabetes Treatment Guide 2010 by the Japan Diabetes Society, HbA1c (NGSP) ≥7.4% was defined as "poor" control in diabetic patients.
The average HbA1c of type 1 diabetic patients in Japan in that era was 8.2-8.3% in a report that included 793 patients in 2006 and 7.8% in a report of 2009, even under the control of a diabetic specialist. According to another Japanese report, even the induction of continuous subcutaneous insulin infusion therapy (CSII), which is an advanced treatment method for type 1 diabetes, limited HbA1c improvement was shown, with the value improving from 9.1% before induction to 8.2% at 6 months and 8.3% at 1 year after the induction of CSII.
In that era, HbA1c (JDS) was used in Japan, and the above description is a revised value as HbA1c (JDS) +0.4 = HbA1c (NGSP); this revision makes the data from that era compatible with the international standard (NGSP). In Japan, where there was a severe shortage of donors, it was assumed that there would be a high need for insulin, even after islet transplantation, unlike in the North American region where insulin withdrawal can be achieved following multiple transplants in a short period. When aiming for HbA1c ≤7.0% in type 1 diabetic patients with an insufficient islet graft volume, it has been assumed that the risk of hypoglycemia was higher (equivalent to level 2), and that priority of safety of Japanese islet recipients should have been considered.
As mentioned above, considering the survey of Japanese type 1 diabetic patients managed by a diabetic specialist, which was used for comparison, and the control of diabetes management goals and islet transplant recipients in Japan, the results of intervention studies for patients who received CSII, and the peculiarity of the transplantation environment in Japan, which has a severe shortage of donors, HbA1c (NGSP) <7.4% was set as the primary endpoint for determining the efficacy and safety of islet transplantation treatment.
Comment 3
page 8 line 315 : what were the criteria for insulin independence ?
Response:
The definition of insulin withdrawal was as follows:
Insulin withdrawal is defined by the criteria, which must be met within 7 days without the administration of insulin.
(1) HbA1c <7.4% or 2.5% decrease from the pre-transplant value
(2) Perform blood glucose measurement (capillary blood is acceptable) at least 4 times or more, before breakfast and 2 hours after each meal per day
Out of these measurements,
Fasting blood glucose level (capillary blood): ≥140 mg/dl; ≤3 times
Blood glucose level: ≥180 mg/dl at 2 hours after the meal; ≤3 times
(3) Fasting blood glucose: ≤126 mg/dl
(4) Fasting C-peptide or C-peptide after stimulation: ≥10.5ng/ml

Round 2
Reviewer 1 Report
The authors have revised their manuscript. Although the revised manuscript is much improved, there are some minor issues and comments that will needed to be fine-tuned, but otherwise supporting publication of this very nice work.
Comment 1
“Before the amendment of the law, brain death was limited to human death only in the case of donation for organ transplantation” This statement makes the readers wonder if a legal definition of brain death (or limitation of brain death) was modified in the amended law. Was a legal definition of brain death modified after the amendment? If so, please describe details. If not, please modify or delete the statement above.
Comment 2
The authors are right in that they never state that the inferior outcomes of PTA and PAK are the only reason. But the authors state “With the categories of pancreas transplantation after kidney and pancreas transplantation alone being associated with worse graft survival in Japan, islet transplantation has become a promising treatment for patients with type 1 diabetes and this is an additional reason to use pancreatic grafts from brain-dead donors for islet transplantation”. This statement clearly indicates that the inferior outcomes of PTA and PAK are an additional reason for using brain-dead donors for islet transplantation, which is not logical. The authors should modify the statement above.
Comment 3
The reviewer’s comment during the initial review was on the authors’ statement that the Working Group of Islet Transplantation has conducted feasibility studies. What are the findings from the feasibility study conducted by the Working Group of Islet Transplantation?
"Guidelines for islet transplantation", "Islet transplantation", "Donation of pancreatic graft for islet transplantation", "Outline of islet transplantation", "Agreement for islet transplantation" and "Manual for clinical islet transplantation in Japan": all may be useful documentations in Japan, but not useful to the international readers. The readers are left uninformed as to what these documentations exactly are. For example, I have no idea what “AGREEMENT” for islet transplantation means. What are differences in contents among "Guidelines for islet transplantation", "Islet transplantation", "Outline of islet transplantation" and "Manual for clinical islet transplantation in Japan"? My recommendation is to summarize what the Working Group of Islet Transplantation conducted rather than to list these titles.
Comments 4-6
The authors addressed my concerns.
Comment 7
The authors should state that the Japanese government recommended using mammalian free (or animal free) enzyme, so that the readers understand why there was no clinical activity in Japan using an alternative enzyme such as Serva enzyme.
Comments 8 - 12
The authors addressed my concerns.
Comment 13
The authors’ response is unclear. The authors cannot explain why islet transplantation is classified as type 1 but bone marrow transplantation is not. Then, I would delete “because it involves the transplantation of cells from another person (donor)”.
Comments 14 and 15
The authors addressed my concerns.
Comment 16
I think the best way is that articles (and websites) written in Japanese should be indicated as such on the reference list.
Please check names: Junko, O.; Takehide, A.; Kazuhiko, J.; Michihiro, M.; Hideaki, M.; Naotake, A.; Seiji, A. I suspect mixed up between first name and last name.
Comments 17 - 32
The authors addressed my concerns.
Author Response
We wish to express our strong appreciation to the reviewer for their insightful comments on our paper. We believe that the comments have helped us to significantly improve the paper.
RESPONSE TO REVIEWER 1
Comment 1
“Before the amendment of the law, brain death was limited to human death only in the case of donation for organ transplantation” This statement makes the readers wonder if a legal definition of brain death (or limitation of brain death) was modified in the amended law. Was a legal definition of brain death modified after the amendment? If so, please describe details. If not, please modify or delete the statement above.
Response:
This following part has been modified (P.2, L.96):
“In Japan, legal brain death is limited to human death only in the case of donation for organ transplantation. Before the amendment of the law, since brain-dead organ donation required the person to make their intention clear before brain death, for the first 10 years (approximately) only a few cases of brain-dead organ transplantation were performed each year.”
Comment 2
The authors are right in that they never state that the inferior outcomes of PTA and PAK are the only reason. But the authors state “With the categories of pancreas transplantation after kidney and pancreas transplantation alone being associated with worse graft survival in Japan, islet transplantation has become a promising treatment for patients with type 1 diabetes and this is an additional reason to use pancreatic grafts from brain-dead donors for islet transplantation”. This statement clearly indicates that the inferior outcomes of PTA and PAK are an additional reason for using brain-dead donors for islet transplantation, which is not logical. The authors should modify the statement above.
Response:
The following part has been deleted from section 2 (P.3):
“and this is an additional reason to use pancreatic grafts from brain-dead donors for islet transplantation”
Comment 3
The reviewer’s comment during the initial review was on the authors’ statement that the Working Group of Islet Transplantation has conducted feasibility studies. What are the findings from the feasibility study conducted by the Working Group of Islet Transplantation?
"Guidelines for islet transplantation", "Islet transplantation", "Donation of pancreatic graft for islet transplantation", "Outline of islet transplantation", "Agreement for islet transplantation" and "Manual for clinical islet transplantation in Japan": all may be useful documentations in Japan, but not useful to the international readers. The readers are left uninformed as to what these documentations exactly are. For example, I have no idea what “AGREEMENT” for islet transplantation means. What are differences in contents among "Guidelines for islet transplantation", "Islet transplantation", "Outline of islet transplantation" and "Manual for clinical islet transplantation in Japan"? My recommendation is to summarize what the Working Group of Islet Transplantation conducted rather than to list these titles.
Response:
The text has been changed below (P.3, L.130):
“This working group also published several documents for the purpose of the achievement of safe and effective clinical islet transplantation, including guidelines and manuals [31]. These documents were eventually summarized in the Manual for Clinical Islet Transplantation in Japan, 4th edition, which was published at the time when islet transplantation received national health insurance coverage in 2020 [32].”
Comment 7
The authors should state that the Japanese government recommended using mammalian free (or animal free) enzyme, so that the readers understand why there was no clinical activity in Japan using an alternative enzyme such as Serva enzyme.
Response:
The following text has been added (P.7, L.319 ):
“At that time, the use of enzymes other than Liberase for islet isolation (e.g., Serva enzyme), was also considered in Japan. However, clinical islet transplantation was performed as part of a single-center study at that time. It was not possible to carry out clinical islet transplantation using these which was due to their higher cost.”
Comment 13
The authors’ response is unclear. The authors cannot explain why islet transplantation is classified as type 1 but bone marrow transplantation is not. Then, I would delete “because it involves the transplantation of cells from another person (donor)”.
Response:
Part of the sentence “because it involves the transplantation of cells from another person (donor)” has been deleted. (P.9, L.)
Comment 16
I think the best way is that articles (and websites) written in Japanese should be indicated as such on the reference list.
Please check names: Junko, O.; Takehide, A.; Kazuhiko, J.; Michihiro, M.; Hideaki, M.; Naotake, A.; Seiji, A. I suspect mixed up between first name and last name.
Response:
Reference 31 has been modified as follows:
- Kenmochi, T.; Ono, J.; Asano, T.; Jingu, K.; Maruyama, M.; Miyauchi, H.; Akutsu, N.; Arita, S.; Iwashita, C., Manual for Clinical Islet Transplantation in Japan.